# Postsynaptic mitochondria are positioned to support functional diversity of dendritic spines

**Connon I Thomas[1], Melissa A Ryan[1†], Naomi Kamasawa[1], Benjamin Scholl[2*‡]**

[1]Electron Microscopy Core Facility, Max Planck Florida Institute for Neuroscience, Max Planck Way, Jupiter, United States; [2]Department of Neuroscience, Perelman School of Medicine at the University of Pennsylvania, Philadelphia, United States

**Abstract** Postsynaptic mitochondria are critical for the development, plasticity, and maintenance of synaptic inputs. However, their relationship to synaptic structure and functional activity is unknown. We examined a correlative dataset from ferret visual cortex with in vivo two-photon calcium imaging of dendritic spines during visual stimulation and electron microscopy reconstructions of spine ultrastructure, investigating mitochondrial abundance near functionally and structurally characterized spines. Surprisingly, we found no correlation to structural measures of synaptic strength. Instead, we found that mitochondria are positioned near spines with orientation preferences that are dissimilar to the somatic preference. Additionally, we found that mitochondria are positioned near groups of spines with heterogeneous orientation preferences. For a subset of spines with a mitochondrion in the head or neck, synapses were larger and exhibited greater selectivity to visual stimuli than those without a mitochondrion. Our data suggest mitochondria are not necessarily positioned to support the energy needs of strong spines, but rather support the structurally and functionally diverse inputs innervating the basal dendrites of cortical neurons.

**\*For correspondence:**
benjamin.scholl@pennmedicine.upenn.edu

**Present address:** †Department of Neuroscience, Baylor College of Medicine, Houston, United States; ‡Department of Physiology and Biophysics, University of Colorado School of Medicine, Aurora, United States

**Competing interest:** The authors declare that no competing interests exist.

## eLife assessment

This study reports **valuable** findings on the correlation between the positions of dendritic mitochondria and the orientation preference of calcium responses of individual spines. The conclusion about the biased localization of dendritic mitochondria near functional diverse spines is informative to understand the functions of dendritic mitochondria. The experimental evidence supporting the conclusion is **compelling**.

## Introduction

By volume, mitochondria are the most prolific organelle in neuronal compartments. Mitochondria are observed within cell bodies, dendritic processes, axons, and occasionally dendritic protrusions (**Cameron et al., 1991**; **Li et al., 2004**; **Kasthuri et al., 2015**; **Faitg et al., 2021**; **Pekkurnaz and Wang, 2022**). Their abundance is thought to reflect a critical role: supporting the energy demands of neuronal communication through a supply of adenosine triphosphate (ATP) (**Datta and Jaiswal, 2021**). Neuronal communication occurs through synapses, and the process of transforming action potentials into graded postsynaptic depolarization is one of the most energy-expensive processes in the brain (**Rossi and Pekkurnaz, 2019**). ATP supplied by mitochondria is needed for postsynaptic glutamate receptor activation, presynaptic vesicle endo/exocytosis pathways, glutamate recycling, and $Ca^{2+}$ channel activity (**Attwell and Laughlin, 2001**; **Harris et al., 2012**). Mitochondria are also involved in local protein synthesis, protein transport, and $Ca^{2+}$ buffering critical for synapse

maintenance and structural dynamics (*Lipton and Whittingham, 1982*; *Li et al., 2004*; *Cheng et al., 2010*; *Harris et al., 2012*; *Rangaraju et al., 2014*; *Datta and Jaiswal, 2021*). While there is growing evidence that mitochondria support synaptic activity, how this support is achieved is largely unknown. Some research suggests that ATP might diffuse from distant sources (*Pathak et al., 2015*), but the limited rate of diffusion of ATP in the cytosol implies the importance of spatial positioning close to synaptic compartments (*Belles et al., 1987*; *Hubley et al., 1996*; *Devine and Kittler, 2018*). Thus, it is unclear to what extent mitochondrial positioning is orchestrated within neuronal compartments to meet the demands of local synaptic inputs.

Adding further complexity, mitochondria are heterogeneous in morphology and density, in a manner dependent on subcellular compartment (e.g. presynapse vs. postsynapse). Presynaptic mitochondria are generally smaller, shorter, and less complex than those in dendrites of the same brain area (*Delgado et al., 2019*; *Faitg et al., 2021*; *Turner et al., 2022*). Ultrastructural correlations reveal a strong link between synaptic strength and mitochondrial abundance: larger boutons, which typically contain a greater number of presynaptic vesicles and are correlated with synapse strength and efficacy (*Schikorski and Stevens, 1997*), contain larger mitochondria and, occasionally, a greater number of mitochondria (*Wilke et al., 2013*; *Cserép et al., 2018*; *Rodriguez-Moreno et al., 2018*). Scaling to meet presynaptic energy demands has also been shown in the calyx of Held where synapse functional maturation correlates with a dramatic increase in mitochondrial volume and organization (*Thomas et al., 2019*). However, while evidence grows for the specific recruitment of mitochondria to the presynapse, the nature of *postsynaptic* mitochondria is largely unstudied (*Faitg et al., 2021*).

Postsynaptic mitochondria are primarily located within dendritic shafts, rarely found extending into dendritic spines or inside the spine head (*Cameron et al., 1991*; *Li et al., 2004*; *Kasthuri et al., 2015*). Dendritic mitochondria are largely thought to provide local energetic support, based on studies relating mitochondrial and synapse abundance. Mitochondrial density along the dendrite co-varies with synapse density in the neocortex (*Turner et al., 2022*). Additionally, dendritic mitochondria are found closer to excitatory synapses than expected by chance in retinal ganglion cells, and a similar relationship has been shown in cultured neurons (*Li et al., 2004*; *Chang et al., 2006*; *Faits et al., 2016*). In vivo measurements of mitochondria $Ca^{2+}$ influx correlate with cytosolic calcium influx (*Datta and Jaiswal, 2021*), potentially supporting local synaptic activity via calcium buffering. However, a missing piece of this framework is whether dendritic mitochondria are organized to support the structure and function of individual dendritic spines and their synaptic activity.

A spatial relationship between mitochondria and dendritic spines may be critically important. Dendritic spines exhibit a wide diversity of morphologies and functional response properties compared to the somatic output (*Scholl et al., 2021*). Given the wide collection of synaptic inputs, conveying distinct information (*Scholl and Fitzpatrick, 2020*) at different input rates (*Goris et al., 2015*), it is reasonable to suggest that mitochondria are precisely positioned to support highly active synapses onto dendritic spines or those with greater influence on somatic action potentials. In addition, dendrites can exhibit homogeneous or heterogeneous clustering of synaptic activity (*Wilson et al., 2016*), and functional organization motifs may require distinct energetic support. However, it is an open question whether mitochondria are preferentially positioned to support certain populations of dendritic spines over others based on their structure or sensory-driven activity.

We address this question by combining in vivo two-photon $Ca^{2+}$ imaging of dendritic spines driven by visual stimuli and serial block-face scanning electron microscopy (SBF-SEM) reconstructions of L2/3 pyramidal neurons of ferret visual cortex (*Scholl et al., 2021*). Using a deep learning method for unsupervised 3D identification of mitochondria in the EM volumes, we characterized the volume of dendritic mitochondria near individual dendritic spines and compared it to structural and functional measurements. While mitochondrial volume was unrelated to dendritic spine morphology, we discovered a surprising functional correlation: local mitochondria were positioned near spines differentially tuned to the somatic output and those in regions with functional heterogeneity. Additionally, spines that contain a mitochondrion in the head or neck were larger and more selective to visual stimuli. Our novel dataset and analysis suggest that dendritic mitochondria are precisely positioned to support functional synaptic diversity of cortical neurons.

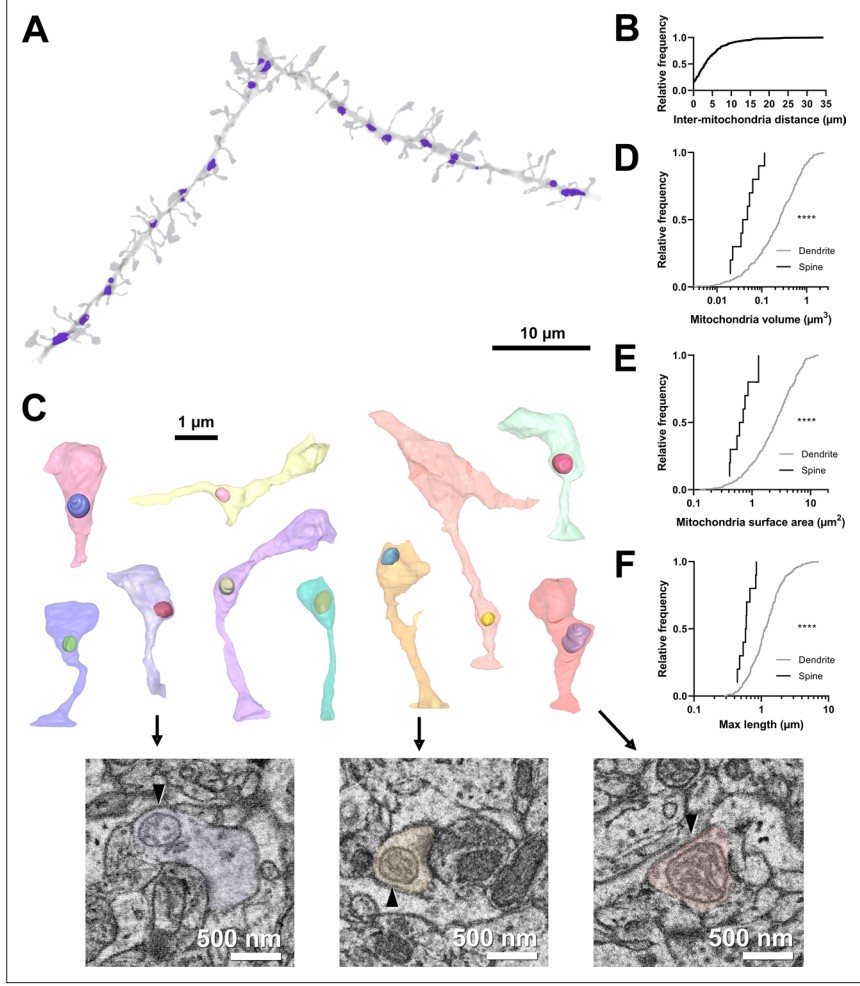

**Figure 1.** Volumetric characterization of postsynaptic mitochondria in L2/3 pyramidal neurons of ferret visual cortex. (**A**) Example electron microscopy (EM) reconstruction of a dendrite with mitochondria (purple). (**B**) Cumulative frequency distribution of inter-mitochondria distance. (**C**) EM reconstructions of all functionally characterized spines that contain mitochondria in the head or neck of the spine. Mitochondria were observed either fully within the head, at the base of the head near the opening of the neck, or in the neck. Electron micrographs show cross sections through three example spines having mitochondria (arrowheads) at the base of the head (purple), fully within the head (yellow), or in the neck (pink). (**D**) Cumulative frequency distribution of mitochondria volume, comparing mitochondria found in dendrites (gray) versus those found in spines (black). (**E**) Mitochondria surface area. (**F**) Maximum mitochondria length. Mann-Whitney $U$ tests; ****p<0.0001.

## Results

### Characterizing postsynaptic mitochondria in 3D EM volumes from ferret visual cortex

To investigate the subcellular organization of dendritic mitochondria, we returned to our previously published CLEM dataset (*Scholl et al., 2021*). This dataset contains individual layer 2/3 neurons from the ferret visual cortex for which basal dendrites were examined in vivo with two-photon calcium imaging of GCaMP6s during visual stimulation, and then subsequently recovered for SBF-SEM and volumetric 3D reconstruction to characterize ultrastructural anatomy. In this study, we continued to extract ultrastructural information from the dendrites imaged in vivo, focusing on postsynaptic mitochondria. To identify postsynaptic mitochondria, we used an unsupervised method for 3D identification of mitochondria in the EM volumes (see Methods). This method quickly and accurately identified mitochondria within the dendritic shaft (*Figure 1A*). From these

3D annotations, we first characterized the basic structure and organization of postsynaptic mito-chondria in our dataset (n=324 mitochondria identified within 63 dendritic segments from 4 cells collected from 2 animals).

Mitochondria were distributed throughout dendrites, with infrequent, large (>10 µm) gaps between individual mitochondria (median inter-mitochondrion distance = 3.025 µm, IQR = 5.281 µm, *Figure 1A and B*). Mitochondria in the basal dendrites of ferret layer 2/3 cortical cells were typically small (median volume = 0.248 µm³, IQR = 0.443 µm³, surface area = 2.537 µm², IQR = 3.144 µm², *Figure 1D–E*), with a short length through the dendrite (maximum length median = 1.137 µm, IQR = 0.881 µm, *Figure 1F*), in comparison to many in vitro and in vivo reports of mouse cortical neurons (*Lewis et al., 2018*). While most mitochondria were positioned within the dendritic shaft, we did observe a small fraction of mitochondrion positioned either inside the spine neck or head (4.9%, 10/203 spines, *Figure 1C*). Of these spines, mitochondria were found entirely within the spine head (2/10), at the junction between the head and neck (5/10), or within the neck (3/10). Mitochondria in spines were significantly smaller and more spherical in shape than dendritic mitochondria (median volume = 0.050 µm³, IQR = 0.047 µm³, surface area = 0.658 µm², IQR = 0.540 µm², maximum length = 0.590 µm, IQR = 0.249 µm; all p<0.0001, Mann-Whitney *U* tests; *Figure 1D–F*). These mitochondria were never continuations of dendritic mitochondria.

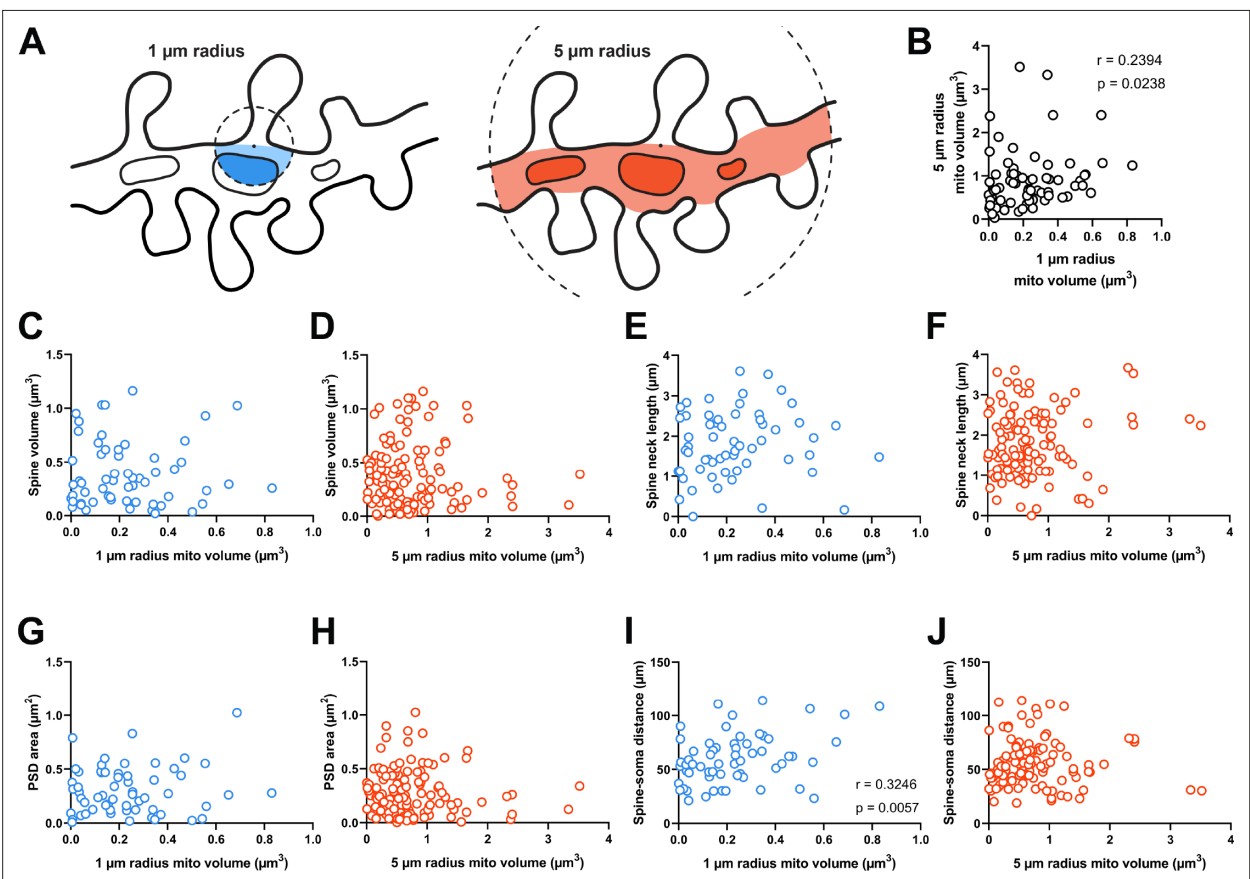

**Figure 2.** Dendritic mitochondria are positioned to support a diversity of anatomical strengths. (**A**) Schematic illustration of the method used to measure dendritic mitochondria volume near structurally and functionally characterized spines in electron microscopy (EM) reconstructions. A sphere of a short (1 µm, blue) or extended radius (5 µm, orange) from the base of the spine neck (black dot) was used to sample the nearby volume of mitochondria for each spine. (**B**) Correlation plot of the volume of mitochondria present within short and extended radii for individual spines. Only spines that had dendritic mitochondria within 5 µm were considered. (**C, D**) Correlation of spine volume and mitochondria volume within a short or extended distance. (**E, F**) Correlation of spine neck length and mitochondria volume. (**G, H**) Correlation of postsynaptic density (PSD) area and mitochondria volume. (**I, J**) Correlation of the distance between a spine/soma and mitochondria volume. *r*=Spearman's correlation coefficient. Correlation plots exclude spines that have zero mitochondria volume within the corresponding measurement radius.

# Dendritic mitochondria organization is independent of dendritic spine morphology

We hypothesized that dendritic mitochondria are strategically positioned to support synapses on dendritic spines in a structural- or activity-dependent manner, as observed for axons (*Smith et al., 2016*). To test this hypothesis, we measured the total mitochondria volume within a given distance from the base of the spine neck (*Figure 2A*). We chose this metric over measuring distance to the nearest mitochondrion for three reasons: (1) distance to a mitochondrion is ill defined and could be measured from either the center of mass or nearest membrane, (2) larger mitochondria produce more ATP and provide a greater capacity for calcium buffering to support nearby synapses, and (3) multiple mitochondria were occasionally clustered nearby individual spines. In presynaptic boutons, a critical inflection-point distance has been identified, where the presence of a mitochondrion less than 3 μm from the synapse results in an increase in active zone size and docked vesicle number (*Smith et al., 2016*). Based on this observation, we measured mitochondria volume using two radii: 1 μm and 5 μm (*Figure 2A*). In our dataset, the median spine neck length is roughly 1.8 μm; therefore, 1 μm and 5 μm distances represent short and extended ranges relative to the synapse. In our population, 56.8% of spines had no mitochondria volume within 1 μm and 12.1% of spines had none within 5 μm. Expectedly, there was greater mitochondria volume for a 5 μm radius compared to 1 μm nearby individual spines (1 μm radius median volume = 0.221 μm³, IQR = 0.278 μm³; 5 μm radius median volume = 0.771 μm³, IQR = 0.561 μm³; n=69 spines, p<0.001, Mann-Whitney *U* test), although these two measures were significantly correlated (Spearman's *r*=0.239, p=0.0238; *Figure 2B*). For all subsequent analyses unless otherwise stated, we focused *only* on spines with measurable mitochondrial volume within each radius.

Dendritic spines differ in shape and size (e.g. *Figure 1B*). As spine structure is highly correlated with synaptic strength (*Bartol et al., 2015*; *Holler et al., 2021*), we examined whether mitochondria were preferentially located nearby stronger or weaker synapses, focusing on spines without a mitochondrion in the head or neck. Spine head volume and postsynaptic density (PSD) area, features that are positively correlated with synapse strength, were uncorrelated with mitochondria volume

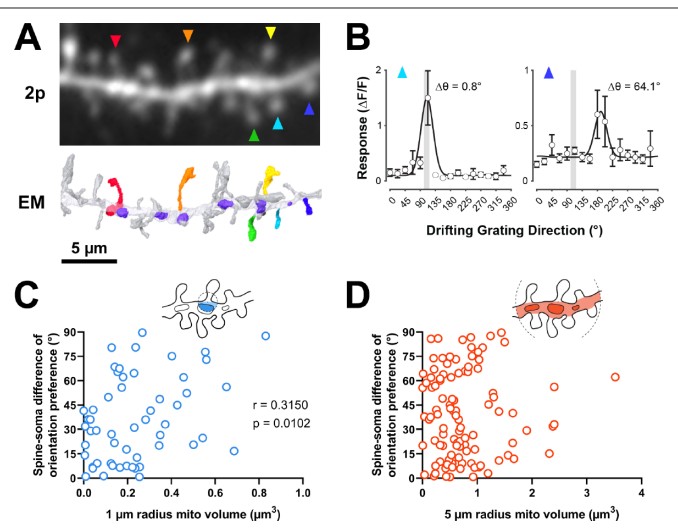

**Figure 3.** Dendritic mitochondria are positioned to support spines with a diversity of orientation tuning. (**A**) Two-photon average projection image of a short segment of GCaMP6s-labeled dendrite and the corresponding electron microscopy (EM) reconstruction. Functionally characterized spines are indicated with rainbow colors/arrows, mitochondria in the EM reconstruction are purple. (**B**) Peak ΔF/F responses in response to visual stimulus for two example spines shown in **A** (teal and indigo), showing two different orientation preferences. The teal spine has a preference that is aligned with the average preference of the soma (vertical gray bar), while the indigo spine has a preference that is dissimilar to the soma. Spine data are mean ± s.e.m. (n=8 stimulus trials). (**C, D**) Correlation of spine-soma difference of orientation preference and mitochondria volume for short (1 μm, blue) and extended (5 μm radius, orange) distances from the base of a spine. *r*=Spearman's correlation coefficient. Correlation plots exclude spines that have zero mitochondria volume within the corresponding measurement radius.

in a 1 µm radius (spine head volume: $r=-0.032$, p=0.403, n=60; PSD area: $r=0.066$, p=0.309, n=60 spines; *Figure 2C and G*) or a 5 µm radius (spine head volume: $r=-0.034$, p=0.357, n=121; PSD area: $r=-0.072$, p=0.216, n=121 spines; *Figure 2D and H*). Spine neck length was also uncorrelated with mitochondria volume at both radii (1 µm: $r=0.185$, p=0.079, n=60 spines; 5 µm: $r=0.040$, p=0.333, n=121 spines; *Figure 2E and F*). We did observe a positive correlation between local 1 µm radius mitochondria volume and the distance of a dendritic spine from the cell body ($r=0.325$, p=0.006, n=60 spines; *Figure 2I*). While this relationship was not observed for mitochondria within a 5 µm radius (5 µm: $r=0.104$, p=0.129, n=121 spines; *Figure 2J*), there is a simple explanation: the volume fraction of dendrite occupied by mitochondria increases with distance from the soma (*Turner et al., 2022*). Dendrites are thicker near the soma and possess few dendritic spines.

## Dendritic mitochondria are positioned near dendritic spines with diverse visual response properties

As dendritic mitochondria position was unrelated to anatomy, we next considered the visually driven calcium activity of individual dendritic spines. In ferret V1, neurons and dendritic spines have exquisite response specificity to the orientation of drifting gratings (*Wilson et al., 2016*). Somatic orientation tuning is thought to arise from the co-activation of inputs sharing similar tuning (*Scholl et al., 2021*). Given the importance of co-tuned inputs supporting somatic selectivity, we hypothesized that dendritic mitochondria would be strategically positioned for energetic support and maintenance of co-tuned inputs.

To test this hypothesis, we examined spines in our EM volumes which were also captured by in vivo two-photon calcium imaging during the presentation of oriented drifting gratings (*Figure 3A*). As previously described (*Scholl et al., 2021*), evoked calcium signals were measured to characterize the orientation tuning of individual dendritic spines and the corresponding somatic output (*Figure 3B*). Each of the four cells in our dataset was similarly tuned for orientation (selectivity range = 0.40–0.57, mean = 0.46 ± 0.08 s.d.). For each tuned spine (selectivity >0.1, n=148/164 spines; see Methods), we calculated orientation preference and compared this value to the orientation tuning of each parent soma (ΔΘpref; see Methods). As shown in *Figure 3B*, spines can be co-tuned with the soma or differentially tuned. We found that mitochondrial volume within a 1 µm radius was correlated with difference in orientation preference relative to the soma (bootstrapped spearman's $r=0.315 ± 0.11$ s.e.m., p=0.010, n=54 spines; *Figure 3C*). This relationship could not be attributed to different properties of co-tuned (ΔΘpref <22.5°, n=68) and differentially tuned spines (ΔΘpref >67.5°, n=30) (selectivity: p=0.12, max response amplitude: p=0.58). When increasing the radius to 5 µm, however, no correlation was observed (bootstrapped Spearman's $r=0.017 ± 0.10$ s.e.m., p=0.432, n=107 spines; *Figure 3D*). This trend was found for all four cells examined (1 µm: mean Spearman's $r=0.49 ± 0.22$ s.d.; 5 µm: mean Spearman's $r=0.09 ± 0.13$ s.d.). We also considered other spine response properties related to tuning preference; specifically, orientation selectivity and response amplitude at the preferred orientation. For either metric, we observed no relationship to mitochondria within 1 µm radius (selectivity: 1 µm: Spearman's $r=-0.081$, p=0.269, n=60; max response amplitude: 1 µm: Spearman's $r=-0.179$, p=0.078, n=64) but did see a weak, significant relationship to both at a 5 µm radius (selectivity: Spearman's $r=0.175$, p=0.027, n=121; max response amplitude: Spearman's $r=-0.166$, p=0.030, n=129). This weak relationship suggests that within some dendritic segments, a greater volume of mitochondria leads to rapid removal of calcium out of the spine and cytoplasm, in agreement with a recent modeling study (*Leung et al., 2021*).

We next wondered whether mitochondria might be positioned near clusters of spines with diverse orientation preferences. Within the same dendritic segment, local populations of spines can either share the same orientation (homogeneous; *Scholl et al., 2017*) or exhibit diversity (heterogeneous; *Wilson et al., 2016*). We measured the average orientation difference (i.e. local heterogeneity) of all spines within a 5 µm neighborhood of a given spine (*Figure 4A*; see Methods). We choose 5 µm because this value is strongly associated with the spatial extent of synaptic clustering and provides adequate sample sizes for computing the average orientation differences (n>2 spines). Similar to spine-soma difference in orientation preference, we found that local heterogeneity was strongly correlated with mitochondria volume within a 1 µm radius ($r=0.563$, p=0.0005, n=31 spines; *Figure 4B*), and not within 5 µm ($r=0.017$, p=0.432, n=107 spines; *Figure 4C*). Importantly, while local heterogeneity was correlated with spine co-tuning (circular $r=0.33$, p=0.0024, n=100 spines), this correlation was entirely

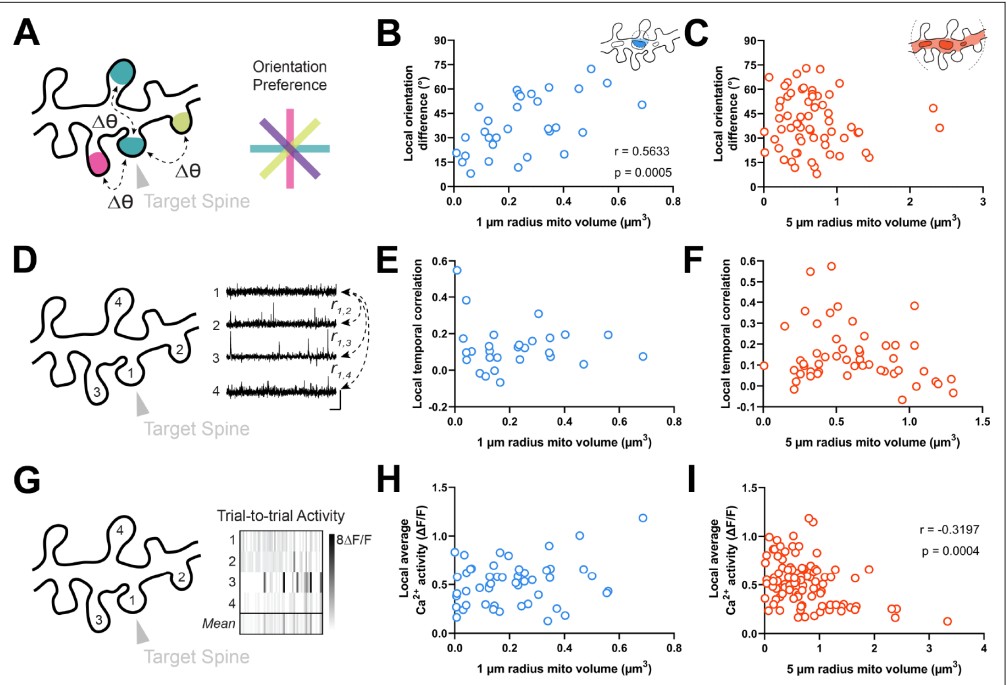

**Figure 4.** Dendritic mitochondria are positioned locally in regions having diverse synaptic orientation preferences and broadly in areas of low local Ca²⁺ activity. (**A**) Illustration of average difference in orientation preference between a target spine (arrowhead) and neighboring spines within 5 µm. (**B, C**) Correlation of local orientation difference and mitochondria volume for short (1 µm, blue) and extended (5 µm radius, orange) distances form the base of a spine. (**D**) Illustration of average local temporal correlation of spines within 5 µm of the target spine. Scale bar is 1 s and 100% ΔF/F. (**E, F**) Correlation of local temporal correlation and mitochondria volume. (**G**) Illustration of local Ca²⁺ activity, shown as mean trial-to-trial ΔF/F, within 5 µm of the target spine. (**H, I**) Correlation of local average Ca²⁺ activity and mitochondria volume. *r*=Spearman's correlation coefficient. Correlation plots exclude spines that have zero mitochondria volume within the corresponding measurement radius.

due to inputs that closely matched the somatic preference. Restricting to only differentially tuned inputs (ΔΘpref >11.25°) yielded no correlation between spine-soma similarity and local heterogeneity (circular *r*=0.03, p=0.82, n=71 spines). Focusing on these spines, local heterogeneity was still strongly correlated with mitochondria volume within 1 µm (Spearman's *r*=0.52, p=0.0052, n=50).

Our data show that dendritic mitochondria are not observed near co-tuned spines or homogeneous clusters, but these measurements are based on orientation tuning preference, a metric that does not necessarily reflect local synaptic activity. Thus, we wondered whether a similar relationship exists for correlated activity between spines or the total calcium activity from groups of spines. As functional similarity coincides with spine co-activity (*Scholl et al., 2017*), dendritic mitochondria should not be positioned near spines with greater co-activity or total activity. We examined both the average local temporal correlation of calcium activity (*Figure 4D*) and trial-by-trial average calcium activity for all spines within a 5 µm radius of a given spine (*Figure 4G*; see Methods). Local temporal correlation was calculated using the average Pearson correlation coefficient of time-varying ΔF/F traces between a target spine and all neighboring spines within 5 µm. We found no relationship for either measurement within a 1 µm radius (local temporal correlation: *r*=0.015, p=0.469, n=28 spines; local average calcium activity: *r*=0.160, p=0.131, n=51 spines; *Figure 4E and H*), nor for local temporal correlation at a 5 µm radius (*r*=–0.152, p=0.141, n=52 spines; *Figure 4F*). We did, however, observe a considerable negative correlation between mitochondria volume within a 5 µm radius and local average calcium activity (*r*=–0.320, p=0.0004, n=105 spines; *Figure 4I*). As this correlation was only observed at an extended radius, it suggests that dendritic regions with a particularly large volume of mitochondria, calcium is quickly buffered and calcium activity is attenuated within dendritic spines.

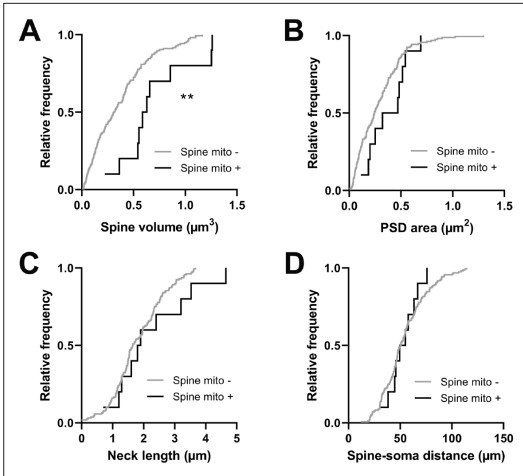

**Figure 5.** Spines having a mitochondrion in the head or neck are larger. (**A**) Cumulative frequency distribution of spine head volume for spines that have a mitochondrion in the head or neck (black, n=10 spines, shown in *Figure 1C*) versus spines that do not (gray). (**B**) PSD area. (**C**) Neck length. (**D**) Distance between the spine and soma. Mann-Whitney *U* tests, **p<0.01.

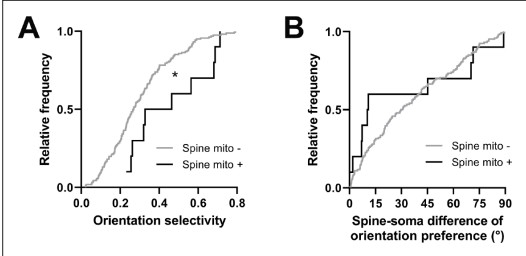

**Figure 6.** Spines having a mitochondrion in the head or neck are more selective for visual features. (**A**) Cumulative frequency distribution of orientation selectivity for spines that have a mitochondrion in the head or neck (black, n=10 spines, shown in *Figure 1C*) versus spines that do not (gray). (**B**) Spine-soma difference of orientation preference. Mann-Whitney *U* tests, *p<0.05.

## Dendritic spines with mitochondria are large and have high tuning selectivity

Our analyses have revealed that dendritic mitochondria are proximal to differentially tuned spines and dendritic regions of local heterogeneity. These analyses have excluded dendritic spines containing mitochondria in the spine neck or head (*Figure 1C*). We then examined whether these unique spines (n=10) displayed distinct structural or functional properties.

Spines with a mitochondrion in the head or neck had a significantly larger spine head volume compared to the majority of spines that do not (mito+: median = 0.608 µm³, n=10 spines; mito–: median = 0.311 µm³, n=157 spines, p=0.0014, Mann-Whitney *U* test, p=0.01, permutation test; *Figure 5A*). A similar trend was observed for PSD area, but it was not significant (mito+: median = 0.396 µm², n=10 spines; mito–: median = 0.244 µm², n=157 spines, p=0.087, Mann-Whitney *U* test, p=0.12, permutation test; *Figure 5B*). Other anatomical properties such as spine neck length (mito+: median = 1.850 µm, n=10 spines; mito–: median = 1.65 µm, n=157 spines, p=0.382, Mann-Whitney *U* test, p=0.14, permutation test; *Figure 5C*) and distance of the spine from the soma (mito+: median = 52.32 µm, n=10 spines; mito–: median = 50.56 µm, n=157 spines, p=0.898, Mann-Whitney *U* test, p=0.42, permutation test; *Figure 5D*) showed no difference between spine groups. Given the difference in size between spines with and without a mitochondrion, we wondered about spines nearby a dendritic mitochondrion (within 1 µm, n=65) compared to those with no measurable mitochondria within 5 µm (n=19). For these groups, we found no difference in spine volume (within 1 µm: median = 0.29 µm³, IQR = 0.41 µm³; no mitochondria within 5 µm: median = 0.40 µm³, IQR = 0.37 µm³; Mann-Whitney ranksum test, p=0.67) or PSD area (within 1 µm: median = 0.26 µm², IQR = 0.33 µm²; no mitochondria within 5 µm: median = 0.31 µm², IQR = 0.36 µm²; Mann-Whitney ranksum test, p=0.49).

Comparing other functional response properties, we found only a single metric showing a significant difference: orientation tuning selectivity. Spines with a mitochondrion were more selective for oriented gratings (mito+: median = 0.396, n=10 spines; mito–: median = 0.268, n=161 spines, p=0.016, Mann-Whitney *U* test, p=0.03, permutation test; *Figure 6A*). Unlike for dendritic mitochondria, we saw no difference in spine-soma difference of orientation preference (mito+: median = 10.62, n=10 spines; mito–: median = 30.17, p=0.451, n=146 spines, Mann-Whitney *U* test, p=0.23, permutation test; *Figure 6B*). Again, we made a similar examination of spines near dendritic mitochondria and those without mitochondria within 5 µm. We found no difference orientation selectivity (1 µm: median = 0.29, IQR = 0.28; no mitochondria within 5 µm: median = 0.28, IQR = 0.15; Mann-Whitney ranksum test, p=0.74) or mismatch to soma orientation (1 µm: median = 0.54°, IQR = 0.86°; no mitochondria within 5 µm: median = 0.46°, IQR = 0.47°; Mann-Whitney ranksum test, p=0.75). Taken together, these data suggest that mitochondria positioned within dendritic spines may serve a different purpose than

those within the dendrites, supporting stronger synaptic inputs, which have been shown to be more selective for features of visual stimuli (*Scholl et al., 2021*).

## Discussion

We used a correlative light and EM dataset (*Scholl et al., 2021*) to characterize mitochondria within basal dendrites of L2/3 pyramidal neurons of the ferret V1 and relate their abundance near spines to an array of ultrastructural and functional properties of those spines. Based on 3D reconstructions using deep learning segmentation, mitochondria were found throughout the dendrites and had a fragmented morphology similar to those observed recently in CA1 pyramidal neuron basal dendrites (*Virga et al., 2023*). We report no relationship to measures of synaptic strength such as spine head volume or PSD area to dendritic mitochondrial volume within either a short or extended (1 μm or 5 μm, respectively) distance from the base of the spine neck. We did observe significant correlations between the volume of nearby mitochondrial and functional diversity – for spine orientation preference similarity to the soma and for local dendritic orientation preference heterogeneity. We also found that spines having mitochondria within the head or neck have significantly larger heads than those that do not and are more selective for visual stimuli, but are no different in other measures of structure and function. Taken together, our results point toward a system where dendritic mitochondria are positioned to support functional diversity, but not synaptic strength as has been previously shown in the presynaptic terminal. Furthermore, though rare, mitochondria within spines do support synaptic strength and selectivity, suggesting that mitochondria are flexible to support different facets of postsynaptic signaling depending on the compartment in which they are present.

### Mitochondria and dendritic spine anatomy

In this study, we investigated whether dendritic mitochondria are preferentially positioned to support certain populations of dendritic spines over others, based on their structural or sensory-driven activity. We were surprised to find that dendritic mitochondria were not positioned near strong synapses, as measured by spine volume or PSD area. Considering that postsynaptic glutamate receptors are estimated to take up 46% of the $O_2$ consumed by oxidative phosphorylation, and that larger spines have greater quantities of receptors (*Holtmaat and Svoboda, 2009*; *Hall et al., 2012*), we assumed that mitochondria would need to be positioned nearby for oxidative support. This is in contrast to studies of the presynapse: presynaptic terminal active zone area and mitochondria volume are strongly correlated, and mitochondria proximity is correlated with the number of nearby docked vesicles (*Smith et al., 2016*; *Thomas et al., 2019*). Our results suggest that dendritic mitochondria are not necessarily positioned to support the energy needs of large spines (i.e. strong synapses) through oxidative phosphorylation, at least in the pyramidal neuron basal dendrites we observed, and opens new questions into the nature of mitochondria.

### Mitochondria and spine functional properties

We hypothesized that dendritic mitochondria would be positioned near synaptic inputs co-tuned with the somatic output (i.e. similar orientation preference). Our rationale was that co-activation of co-tuned inputs are critical for shaping somatic selectivity, and these inputs can often be found to be clustered together to trigger nonlinear dendritic events (*Wilson et al., 2016*; *Scholl et al., 2021*; *Scholl et al., 2017*). Surprisingly, we found the opposite: mitochondria are present in greater volumes near differentially tuned spines. This includes spines that had an orientation preference mismatching the soma and spines within dendritic regions of heterogeneity in orientation preference. What might be the cause of such a striking deviation from our original hypothesis?

One possibility is that regions of high spine functional diversity are undergoing plasticity, such as spine pruning, de novo formation, or strength modifications. Recent in vivo studies have demonstrated that mitochondria are recruited to dendritic regions undergoing structural dynamics (*Faits et al., 2016*; *Dromard et al., 2021*); however, it is unclear whether this is the case for our own EM dataset as we were only able to study a single developmental time point (adolescent, visually mature ferret). Another possibility is that differentially tuned spines and heterogeneous clusters reside nearby a greater number of GABAergic inputs, supported energetically or buffered by local mitochondria, which could thereby act to sculpt away excitatory input. It is also possible that co-tuned spines will

be more co-activated with and infiltrated by back-propagating somatic action potentials (*Palmer and Stuart, 2009*; *Popovic et al., 2015*), as compared to differentially tuned spines. Back-propagating action potentials may alter spine head depolarization, PSD composition, or calcium dynamics; differences in spine activation may lead to different levels of support needed from mitochondria. Finally, it is possible that our observations are specific to the ferret or visual cortex. Unfortunately, there are not enough reported measurements of postsynaptic mitochondria to compare between animal models, let alone brain areas, and it remains to be shown if our results will hold for layer 2/3 pyramidal neurons in mouse V1. Nevertheless, as ferrets are a highly visual animal with a primate-like organization of the early visual system, our observations may reflect specific visual processing demands (e.g. extracting motion information from incoming signals).

Of course, there are limitations precluding our ability to assess dendritic spine structure-function relationships. Visual cortical neurons exhibit a high degree of functional heterogeneity and we probed only a single stimulus dimension (orientation). In vivo spine calcium imaging is biased toward larger spines and can only resolve synaptic events involving NMDA receptors or voltage-gated calcium channels (*Scholl et al., 2021*). Spine calcium signals are likely not to directly reflect underlying electrical signals (*Koester and Johnston, 2005*; *Sobczyk et al., 2005*). In addition, the kinetics of GCaMP6s make it difficult to characterize other functional properties of synapses such as activity frequency during visual stimulation. In summary, although we cannot provide a clear reason for our findings, our data highlight the need for greater investigation into dendritic mitochondria, their subcellular organization within cells, and their relationship to patterns of synaptic activity.

## Mitochondrial structure and distribution

Several factors influence the size and abundance of neuronal mitochondria including brain region, neuronal compartment, cell type, animal age, and health (*Delgado et al., 2019*; *Glancy et al., 2020*). Despite this variability, dendritic mitochondria are generally longer and take up a greater fraction of neurite volume compared to axonal mitochondria, which tend to be more fragmented and, in some cases, more mobile. Previous reports in mouse hippocampal cell culture and rat hippocampus report dendrites with long, overlapping mitochondria that form stable compartments of ~35 μm length or more (*Popov et al., 2005*; *Rangaraju et al., 2019*). Studies of mouse cortex using EM or light microscopy report mitochondria lengths of roughly 1–10 μm (*Lewis et al., 2018*; *Turner et al., 2022*). Our results are closer to those of mouse cortex, but are smaller (median length just over 1 μm, 10th percentile = 0.55 μm, 90th percentile = 2.49 μm). The mitochondrial morphology we observe is most similar to a recent report from mouse CA1 pyramidal neuron basal dendrites, where mitochondria are ~1 μm long and have a similar size distribution (*Virga et al., 2023*). This similarity could be due to the fact that dendritic mitochondrial morphology is cell type and neuronal compartment specific (e.g. apical vs. basal dendrites). Another possibility is that mitochondrial fragmentation, as we observe in layer 2/3 ferret basal dendrites, is linked to synaptic excitability. In support of this hypothesis, *Virga et al., 2023* demonstrated that low levels of excitability and activity is linked to longer mitochondria. Thus, presynaptic neurons driving basal synapses in carnivore V1 may operate at relatively high firing rates (*Mante et al., 2008*), leading to more fragmented mitochondria than compared to mouse V1.

Another variable feature across brain regions is the presence of mitochondria within spines. In hippocampal cell cultures, mitochondria enter about 8–10% of spines, typically as extensions of a dendritic mitochondrion (*Li et al., 2004*). EM reconstructions of mouse neocortex showed only 0.21% of spines (n=3/1425) contained mitochondria (*Kasthuri et al., 2015*). In contrast, for olfactory bulb granule cells, ~87% of spines contain mitochondria (*Cameron et al., 1991*). We found that 4.9% of spines contain mitochondria, however, since we specifically examined layer 2/3 neuron basal dendrites proximal to the soma, it is difficult to ascertain where this value stands within the literature and future studies will need to investigate variations due to brain area, cell type, and compartment. Another interesting distinction in our dataset was that spine mitochondria were roughly spherical or oblong, never continuations of a dendritic mitochondrion. It is likely that these mitochondria are small fragmentations of nearby dendritic mitochondria trafficked into the spine, possibly through interactions with the cytoskeleton (*MacAskill and Kittler, 2010*), supporting the idea that spine mitochondria serve different role than dendritic mitochondria.

Previous observations in cultured neurons suggest that dendritic mitochondria are stable, and similar evidence has been shown in retinal ganglion cell dendrites from the intact retina (*Chang*

*et al., 2006*; *Faits et al., 2016*; *Rangaraju et al., 2019*). Experiments have also shown that oxidative phosphorylation by mitochondria is needed for dendritic protein translation, not glycolysis (*Rangaraju et al., 2019*), and that it contributes a majority of the energy consumed at the postsynaptic site by receptor and ion channel activation (*Harris et al., 2012*). However, taking our observations into account, it's difficult to conceive such a system could exist where mitochondria are simultaneously immobile and provide significant quantities of energy for synaptic activity through oxidative phosphorylation, yet have no positional bias toward large, strong synapses. We expect there exists a balance between oxidative phosphorylation and glycolysis supplying energy at the synapse. This balance could shift depending on brain region, cell type, compartment, activity level, and mitochondrial morphology and mobility. In this way, dendritic areas receiving synaptic inputs with high functional diversity require greater local mitochondrial ATP production, while baseline activity is supported through local glycolysis or ATP diffusion from more distant sources. Additionally, the role that mitochondria serve at the synapse may shift toward one of calcium buffering and plasticity support. Local depletion of mitochondria has been shown to reduce plasticity (*Rangaraju et al., 2019*), and by blocking fission, spines have impaired structural and functional long term potentiation (LTP). Induction of LTP causes increased dendritic mitochondrial $Ca^{2+}$ and a burst of fission (*Divakaruni et al., 2018*). Thus, once spines become established, mitochondria may localize to input-diverse dendritic regions to support synaptic plasticity. More investigation into the support functions of postsynaptic mitochondria is needed to elucidate their different contributions to synaptic function, plasticity, and development.

## Methods

All procedures were performed according to NIH guidelines and approved by the Institutional Animal Care and Use Committee at Max Planck Florida Institute for Neuroscience.

### Animals, viral injections, and cranial windows

Information on animals, survival viral injections, cranial window implantation are described in depth in our previous studies (*Scholl and Fitzpatrick, 2020*; *Scholl et al., 2017*; *Scholl et al., 2021*). Briefly, layer 2/3 neurons of the primary visual cortex of female ferrets (n=3, Marshall Farms) were driven to express the calcium indicator GCaMP6s via a Cre-dependent AAV expression system. On the day of imaging, a cranial window was surgically implanted to image calcium activity of cortical pyramidal neurons during presentation of visual stimuli.

### Two-photon imaging

Two-photon imaging was performed using a Bergamo II microscope (Thorlabs) running Scanimage (Vidrio Technologies) with 940 nm dispersion-compensated excitation provided by an Insight DS+ (Spectraphysics). For spine and axon imaging, power after the objective was limited to <50 mW. Images were collected at 30 Hz using bidirectional scanning with 512×512 pixel resolution or with custom ROIs (regions of interest; framerate range: 22–50 Hz). Somatic imaging was performed with a resolution of 0.488–0.098 µm/pixel. Dendritic spine imaging was performed with a resolution of 0.164–0.065 µm/pixel.

### Visual stimuli

Visual stimuli were generated using Psychopy (*Peirce, 2007*). The monitor was placed 25 cm from the animal. Receptive field locations for each cell were hand mapped and the spatial frequency optimized (range: 0.04–0.20 cpd). For each soma and dendritic segment, square-wave or sine-wave drifting gratings were presented at 22.5° increments to each eye independently (2 s duration, 1 s ISI, 8–10 trials for each field of view). Drifting gratings of different directions (0 – 315°) were presented independently to both eyes with a temporal frequency of 4 Hz.

### Two-photon imaging analysis

Imaging data were excluded from analysis if motion along the z-axis was detected. Dendrite images were corrected for in-plane motion via a 2D cross-correlation-based approach in MATLAB or using a piecewise non-rigid motion correction algorithm (*Pnevmatikakis and Giovannucci, 2017*). ROIs

were drawn in ImageJ; dendritic ROIs spanned contiguous dendritic segments and spine ROIs were fit with custom software. Mean pixel values for ROIs were computed over the imaging time series and imported into MATLAB. ΔF/F was computed with a time-averaged median or percentile filter (10th percentile). We subtracted a scaled version of the dendritic signal to remove back-propagating action potentials as performed previously (*Wilson et al., 2016*). ΔF/F traces were synchronized to stimulus triggers sent from Psychopy and collected by Spike2.

Peak responses to bars and gratings were computed using the Fourier analysis to calculate mean and modulation amplitudes for each stimulus presentation, which were summed together. Spines were included for analysis if the mean peak for the preferred stimulus was >10%, the SNR at the preferred stimulus was >1, and spines were weakly correlated with the dendritic signal (Spearman's correlation, $r<0.4$). Some spine traces contained negative events after subtraction, so correlations were computed ignoring negative values. Preferred orientation for each spine was calculated by fitting responses with a double Gaussian tuning curve (*Wilson et al., 2016*) using lsqcurvefit (MATLAB). Orientation selectivity was computed by calculating the vector strength of mean responses (*Wilson et al., 2016*).

For each tuned spine (orientation selectivity >0.1), we calculated orientation preference difference between the spine and parent soma as in *Wilson et al., 2016*. To measure the average orientation difference (i.e. local heterogeneity), we calculated the orientation preference difference between a target spine and all surrounding spines within 5 µm on the same dendrite. For targets with n>2 neighbors, we then computed the circular mean. Average spatiotemporal correlation for a given spine was calculated similarly: the Pearson correlation between time-varying ΔF/F traces was computed between a target spine and all neighbors within 5 µm, and then we calculated the average of this population. To calculate trial-by-trial average calcium activity, we averaged the amplitudes of evoked calcium transients between a target spine and neighbors within 5 µm for all visual stimulus trials. A grand mean was then computed across all trials.

## Perfusion, fixation, and slice preparation for fluorescence imaging

Anesthetized animals were immediately perfused with 2% paraformaldehyde and 2–2.5% glutaraldehyde in a 0.1 M sodium cacodylate buffer (pH 7.4). Following removal, brains were sliced at 80 µm parallel to the area flattened by the cranial window. Slices were quickly imaged at low magnification (×20, 0.848×0.848 µm/pixel) using a Leica CLSM TCS SP5 II running LAS AF (ver. 3.0, Leica) with 488 nm laser excitation. Fluorescence of GCaMP6s was used to locate the target cell in this view. Autofluorescence resulting from glutaraldehyde fixation also produced signal within the tissue slices. A field of view large enough to cover roughly one-fourth of the slice, with the target cell included, was captured using image tiling. This process was performed to identify the fluorescent cell of interest and for slice-level correlation in later steps of the workflow. The slice containing the target cell was then imaged with the same ×20 objective at higher pixel resolution (0.360×0.360×0.976 µm/voxel) to obtain a z-stack of the full depth of the slice and immediate region surrounding the target cell. CLSM imaging required approximately 1–3 hr to find the cell of interest within a slice and capture a tiled slice overview and higher resolution z-stack.

## Sample preparation for SBF-SEM imaging

Samples were prepared as previously described (*Scholl et al., 2021*; *Thomas et al., 2021*). Briefly, tissue pieces were incubated in an aqueous solution of 2% osmium tetroxide buffered in 0.1 M sodium cacodylate for 45 min at room temperature (RT). Tissue was not rinsed and the osmium solution was replaced with cacodylate buffered 2.5% potassium ferrocyanide for 45 min at RT in the dark. Tissue was rinsed with water 2×10 min, which was repeated between consecutive steps. Tissue was incubated at RT for 20 min in aqueous 1% thiocarbohydrazide dissolved at 60°C, aqueous 1% osmium tetroxide for 45 min at RT, and then 1% uranyl acetate in 25% ethanol for 20 min at RT in the dark. Tissue was rinsed then left in water overnight at 4°C. The following day, tissue was stained with Walton's lead aspartate for 30 min at 60°C. Tissue was then dehydrated in a graded ethanol series (30%, 50%, 70%, 90%, 100%), 1:1 ethanol to acetone, then 100% dry acetone. Tissue was infiltrated using 3:1 acetone to Durcupan resin (Sigma-Aldrich) for 2 hr, 1:1 acetone to resin for 2 hr, and 1:3 acetone to resin overnight, then flat embedded in 100% resin on a glass slide and covered with an Aclar sheet at 60°C for 2 days. The tissue was trimmed to less than 1×1 mm in size, the empty resin

at the surface was shaved to expose tissue surface using an ultramicrotome (UC7, Leica), then turned downward to be remounted to a metal pin with conductive silver epoxy (CircuitWorks, Chemtronics).

## SBF-SEM image acquisition and volume data handling

Samples were imaged using a correlative approach as previously described (*Thomas et al., 2021*). Briefly, tissue was sectioned and imaged using 3View2XP and Digital Micrograph (ver. 3.30.1909.0, Gatan Microscopy Suite) installed on a Gemini SEM300 (Carl Zeiss Microscopy LLC) equipped with an OnPoint BSE detector (Gatan, Inc). The detector magnification was calibrated within SmartSEM imaging software (ver. 6.0, Carl Zeiss Microscopy LLC) and Digital Micrograph with a 500 nm cross line grating standard. A low-magnification image of each block face was manually matched to its corresponding depth in the CLSM Z-stack using blood vessels and cell bodies as fiducials. Final imaging was performed at 2.0–2.2 kV accelerating voltage, 20 µm or 30 µm aperture, working distance of ~5 mm, 0.5–1.2 µs pixel dwell time, 5.7–7 nm per pixel, knife speed of 0.1 mm/s with oscillation, and 56–84 nm section thickness. Acquisition was automated and ranged from several days to several weeks depending on the size of the ROI and imaging conditions. The true section thickness was measured using mitochondria diameter calibrations (*Fiala and Harris, 2001*). Calibration for each block was required, since variation in thickness can occur due to heating, charging from the electron beam, resin polymerization, and tissue fixation and staining quality (*Starborg et al., 2013*; *Hughes et al., 2014*).

Serial tiled images were exported as TIFFs to TrakEM2 (*Schindelin et al., 2012*) within ImageJ (ver. 1.52p) to montage tiles, then aligned using Scale-Invariant Feature Transform image alignment with linear feature correspondences and rigid transformation (*Lowe, 2004*). Once aligned, images were inverted and contrast normalized.

## SBF-SEM image analysis

Aligned images were exported to Microscopy Image Browser (ver. 2.51, 2.6, *Belevich et al., 2016*) for segmentation of dendrites, spines, PSDs, and boutons. Three annotators preformed segmentation and the segmentations of each annotator were proofread by an experienced annotator (~1000 hr of segmentation experience) prior to quantification. Binary labels files were imported to Amira (ver. 6.7, 2019.1, Thermo Fisher Scientific) which was used to create 3D surface models of each dendrite, spine, and PSD. Once reconstructed, the model of each dendrite was manually overlaid onto its corresponding two-photon image using Adobe Photoshop for re-identification of individual spines. Amira was used to measure the volume of spine heads, surface area of PSDs, and spine neck length.

For mitochondrial analyses, segmentations were moved into ORS Dragonfly (ver. 2021.3, 2022.1). Within Dragonfly, a pre-built U-net++ architecture was trained on a small set of manual mitochondria segmentations, then applied for segmentation of all mitochondria within target dendrites. Dendrite segmentations were used as a mask to apply the trained mitochondria network, and mitochondria segmentations were proofread for accuracy. Mitochondria morphology was measured using tools within Dragonfly, while inter-mitochondria distances were measured in Microscopy Image Browser using the measure tool. For quantification of mitochondria volume near spines, a sphere of either 1 µm or 5 µm radius was applied with the center at the base of each spine neck, and the volume of mitochondria intersected by each sphere was measured. Mitochondria within spines were segmented manually and these spines (n=10) were not included in correlation analyses. In addition, analyses excluded spines with 0 µm$^3$ mitochondrial volume within the relevant radii unless otherwise stated.

## Statistics

Statistical analyses are described in the main text and in figure legends. We used non-parametric statistical analyses (Wilcoxon signed-rank test, Mann-Whitney ranksum test) or permutation tests to avoid assumptions about the distributions of the data. Permutation tests used equivalent sample sizes for each distribution and compared data to the null hypothesis. Statistical analysis was performed in MATLAB (2021b) or GraphPad Prism (ver. 9). Circular correlation coefficients were computed for tests with circular variables. For all other tests, Spearman's correlation coefficient was computed. All correlation significance tests were one-sided. Quantitative approaches were not used to determine if the data met the assumptions of the parametric tests.

## Acknowledgements

The authors thank Nicole Shultz and Rachel Satterfield for help with perfusions and fixative preparation, Tal Laviv for comments, the Fitzpatrick lab for useful discussions, and the MPFI ARC for animal care. The authors thank the GENIE project for access to GCaMP6.

## Additional information

### Funding

| Funder | Grant reference number | Author |
|---|---|---|
| Max Planck Florida Institute for Neuroscience | | Connon I Thomas Melissa A Ryan Naomi Kamasawa |
| National Eye Institute | EY031137 | Benjamin Scholl |

The funders had no role in study design, data collection and interpretation, or the decision to submit the work for publication.

### Author contributions

Connon I Thomas, Conceptualization, Data curation, Formal analysis, Investigation, Visualization, Methodology, Writing – original draft, Writing – review and editing; Melissa A Ryan, Conceptualization, Data curation, Methodology, Writing – review and editing; Naomi Kamasawa, Conceptualization, Supervision, Funding acquisition, Writing – review and editing; Benjamin Scholl, Conceptualization, Formal analysis, Supervision, Funding acquisition, Investigation, Writing – original draft, Writing – review and editing

### Author ORCIDs

Connon I Thomas ![ORCID] http://orcid.org/0000-0003-0995-9667
Melissa A Ryan ![ORCID] http://orcid.org/0000-0002-0468-9525
Naomi Kamasawa ![ORCID] http://orcid.org/0000-0002-8926-5309
Benjamin Scholl ![ORCID] https://orcid.org/0000-0002-9578-7234

### Ethics

All procedures were performed according to NIH guidelines and approved by the Institutional Animal Care and Use Committee at Max Planck Florida Institute for Neuroscience (Protocol #23-002).

Reviewer #1 (Public Review): https://doi.org/10.7554/eLife.89682.3.sa1
Reviewer #2 (Public Review): https://doi.org/10.7554/eLife.89682.3.sa2
Reviewer #3 (Public Review): https://doi.org/10.7554/eLife.89682.3.sa3
Author Response https://doi.org/10.7554/eLife.89682.3.sa4

## Additional files

### Supplementary files
• MDAR checklist

### Data availability

Source data for all data figures is publicly available (Zenodo).

The following dataset was generated:

| Author(s) | Year | Dataset title | Dataset URL | Database and Identifier |
|---|---|---|---|---|
| Scholl B | 2023 | Postsynaptic mitochondria are positioned to support functional diversity of dendritic spines | https://zenodo.org/records/10278270 | Zenodo, 10.5281/zenodo.10278270 |

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
