## [Editor Report · eLife assessment]

This study reports **valuable** findings on the correlation between the positions of dendritic mitochondria and the orientation preference of calcium responses of individual spines. The conclusion about the biased localization of dendritic mitochondria near functional diverse spines is informative to understand the functions of dendritic mitochondria. The experimental evidence supporting the conclusion is **compelling**.

---

## [Referee Report · Reviewer #1 (Public Review)]

Summary:

Mitochondria is the power plant of the cells including neurons. Thomas et al. characterized the distribution of mitochondria in dendrites and spines of L2/3 neurons from the ferret visual cortex, for which visually driven calcium responses of individual dendritic spines were examined. The authors analyzed the relationship between the position of mitochondria and the morphology or orientation selectivity of nearby dendrite spines. They found no correlation between mitochondrion location and spine morphological parameters associated with the strength of synapses, but correlation with the spine-somatic difference in orientation preference and local heterogeneity in preferred orientation of nearby spines. Moreover, they reported that the spines that have a mitochondrion in the head or neck are larger in size and have stronger orientation selectivity. Therefore, they proposed that "mitochondria are not necessarily positioned to support the energy needs of strong spines, but rather support the structurally and functionally diverse inputs."

Strengths:

This paper attempted to address a fundamental question: whether the distribution of the mitochondria along the dendrites of visual cortical neurons is associated with the functions of the spines, postsynaptic sites of excitatory synapses. Two state of the art techniques (2 photon Ca imaging of somata and spines and EM reconstructions of cortical pyramidal neurons) had been used on the same neurons, which provides a great opportunity to examine and correlate the functional properties of spine ultrastructure and spatial distribution of dendritic mitochondria. The conclusion that dendritic mitochondria support functional diversity of spines, but not synaptic strength is surprising and will inspire rethinking the role of mitochondria in synaptic functions.

Weaknesses:

Overall, the findings are intriguing. However, the interpretations of these findings need extra cautions due to the limitations of experimental designs and tools in this study. Neurons in L2/3 of visual cortex are highly diverse in functional properties, which is represented by not only orientation selectivity, but also direction selectivity and spatial/temporal frequency selectivity, etc. The orientation tuning with fixed spatial and temporal frequency may not be the optimal way of stimulating individual synaptic inputs to evaluate synaptic strengths. And the correlation between mitochondria distribution and spine activity evoked by other visual stimulation parameters is worth exploration. Moreover, GCaMP6s measures only spine Ca signals mediated by NMDA and voltage-gated Ca channels, but not sodium currents mediated by ligand-gated or voltage-gated channels. Thus, it reports only some aspects of synaptic properties. Future studies with new tools might help resolve those issues.

---

## [Referee Report · Reviewer #2 (Public Review)]

Summary:

Mitochondria in synapses are important to support functional needs, such as local protein translation and calcium buffering. Thus, they may be strategically localized to maximize functional efficiency. In this study, the authors examine whether a correlation exists between the positioning of mitochondria and the structure or function of dendritic spines in the visual cortex of a ferret. Unexpectedly, the authors found no correlation between structural measures of synaptic strength to mitochondria positioning, which may indicate that they are not localized only because of the local energy needs. Instead, the authors discover that mitochondria are positioned preferably in spines that display heterogeneous responses, showing that they are localized to support specific functional needs probably distinct from ATP output.

Strengths:

The thorough analysis provides a yet unprecedented insight into the correlation between synaptic tuning and mitochondrial positioning in the visual cortex in vivo.

Weaknesses:

Analysis of this study suggested that mitochondrial volume does not correlate with structural measures of synaptic strength (e.g. spine volume and post-synaptic density (PSD) area), but it remains to be determined if mitochondria localization is also co-related to the frequency of synaptic activity, and what causes the correlation (driven by mitochondrial positioning, or by synaptic activity).

---

## [Referee Report · Reviewer #3 (Public Review)]

Summary: This is a careful examination of the distribution of mitochondria in the basal dendrites of ferret visual cortex in a previously published volume electron microscopy dataset. The authors report that mitochondria are sparsely, as opposed to continuously distributed in the dendritic shafts, and that they tend to cluster near dendritic spines with heterogeneous orientation selectivity.

Strengths: Volume EM is the gold standard for quantification of organelle morphology. An unusual strength of this particular dataset is that the orientation selectivity of the dendritic spines was measured by calcium imaging prior to EM reconstruction. This allowed the authors to assess how spines with varying selectivity are organized relative to mitochondria, leading to an intriguing observation that they localize to heterogeneous spine clusters. The analysis is carefully performed. An additional strength is the use of a carnivore with a sophisticated visual system.

Weaknesses: Using threshold distances between mitochondria and synapses as opposed to absolute distances may overlook important relationships in the data.

---

## [Author Response]

The following is the authors’ response to the original reviews.

We sincerely thank and express our appreciation to each of the reviewers for their thorough critique of our manuscript.

**Reviewer #1 (Recommendations For The Authors):**
1. The analysis of whole study comes from only 4 cells from L2/3 of ferret visual cortex; however, it is well known that there is a high level of functional heterogeneity within the cortical neurons. Do those four neurons have similar or different response properties? If the four neurons are functionally different, the weak or no correlation may result from heterogenous distribution pattern of mitochondria associated with heterogenous functionality.

This is an important consideration and often a limitation of CLEM studies. While cortical neurons do exhibit a high degree of functional heterogeneity (similar to spine activity), the 4 cells examined all had selective (OSI > 0.4) somatic responses to oriented gratings, although they differed in their exact orientation preference. Due to experimental limitations of recording from a large population of dendritic spines, we did not characterize other response properties for which their sensitivity might differ. We did not consider orientation preference a metric of study, but instead characterized the difference in preference from the somatic output, allowing comparisons across spines. In addition, our measurements were limited to proximal, basal dendrites rather than any location in the dendritic tree. Nonetheless, we attempted to address this concern by examining spines with functionally heterogenous visual responses within single cells, as reported in our manuscript: mitochondrial volume within a 1 µm radius was correlated with difference in orientation preference relative to the soma across all 4 cells, the mean (r = 0.49 +/- 0.22 s.d.), suggesting that cell-to-cell variability has a minimal impact on our main conclusions.

Even with our limited measurements, there is a large amount of functional heterogeneity in dendritic spine responses (Extended Data Figure 2, Scholl et al. 2021), far greater than the small differences in somatic responses of these 4 cells (Figure 3, Scholl et al. 2021). Moreover, the individual dendrites from these 4 cells exhibited substantial heterogeneity in the distribution of mitochondria. We cannot rule out whether heterogeneity at various scales may obscure certain relationships or result in the weak correlations we observed. We also note that future advancements in volume electron microscopy should allow for greater sample sizes that can better address the role of functional (and structural) heterogeneity.We have added text in the Discussion section about the potential structure-function relationships that might be obscured or revealed by neuron heterogeneity.

1. The authors argued that "mitochondria are not necessarily positioned to support the energy needs of strong spines." However, the overall energy needs of a spine depend not only on the synaptic strength but also on the frequency of synaptic activity. Is there a correlation between the mitochondria volume around a spine and the overall activity of the spine? This data needs to be analyzed to confirm the distribution of mitochondria is independent of local energy needs.

The reviewer is correct, but our experimental paradigm was not optimally designed to measure the ‘frequency’ of synaptic activity in vivo. This could have been accomplished with flashed gratings or, perhaps, presenting drifting gratings at different temporal frequencies. For spines tuned to higher temporal frequencies in V1, we may expect greater energy needs, although as the reviewer suggests, energy needs will depend on synapse (and bouton) size. Because we are not able to directly measure activity frequency as carefully or beautifully as can be done ex vivo or in nerve fibers, we do not feel confident in attempting such analysis in this study. Instead, based on previous studies, we assumed that larger synapses might be able to transmit higher frequencies, and thus have higher energy demands. We believe future in vivo studies will more directly measure synaptic frequency for comparison with mitochondria.

We have added text in the Discussion about this caveat and potential future experiments.

1. In the results section, the authors briefly mentioned that "We also considered other spine response properties related to tuning preference; specifically, orientation selectivity and response amplitude at the preferred orientation. For either metric, we observed no relationship to mitochondria within 1 μm radius (selectivity: 1 μm: r = -0.081, p = 0.269, n = 60; max response amplitude: 1 μm: r = -0.179, p = 0.078, n = 64) but did see a weak, significant relationship to both at a 5 μm radius (selectivity: r = 0.175, p = 0.027, n = 121; max response amplitude: r = -0.166, p = 0.030, n = 129)." Here only statistic results were given while the data were not presented in the figure illustration. Based on the methods and Figure 3B, it seems that the preferred orientations were calculated based on the vector summation. How did the authors calculate the "response amplitude at the preferred orientation"? This needs to be clarified. In addition, given the huge variety of orientation selectivity, using the response amplitude at the preferred orientation may not be the best parameter to correlate with the mitochondria volume which is indicative of energy needs. It might be necessary to include the baseline activity without visual stimulation and the average response for visual stimuli of different orientations in the analysis.

We apologize for this oversight, as the details are present in our previous study (Scholl et al., 2021). Response amplitude and preferred orientation were calculated from a Gaussian curve fitting procedure with specific parameters describing those exact values (see Scholl et al. 2021 or Scholl et al. 2013). Only spines with selective responses (vector strength index > 0.1) and passing our SNR criterion were used for these analyses. We have now added this information to the Methods section and referred to it in the Results.With respect to the reviewer’s other concern, we also examined the average response amplitude (across all visual stimuli). There we found no relationship between the volume of mitochondria within 1 or 5 microns of a spine, however, because spines differ greatly in their selectivity (range = 0 – 0.8) the average response may not be an appropriate metric to compare across spines.

1. A continuation from the former point, do the spines with similar preferred orientation to the somatic Ca signal have similar Ca signal strength, orientation selectivity index and other characteristics to the spines with different preferred orientation? As shown in the examples (Figure 3B), the spine on the right with different orientation preference compared with its soma has considerably larger response in non-preferred orientation compared to the spine on the left. Thus, the overall activity of the spine on the right may be higher than the spine which has similar preferred orientation to the soma. The authors showed that a positive correlation between difference in orientation preference and mitochondria volume (Figure 3C). Could this be simply due to higher spine activity for non-preferred orientation or spontaneous activity? Thus, the mitochondria might be positioned to support spines with higher overall activity rather than diverse response property.

The reviewer brings up an interesting consideration. We examined the response properties of spines co-tuned (∆θpref < 22.5 deg) and differentially-tuned (∆θpref > 67.5 deg) to the soma. The orientation selectivity was not different between the two groups (p = 0.12, Wilcoxon ranksum test), although there was a small trend towards co-tuned inputs being more selective. We found that calcium response amplitudes for the preferred stimulus were also not different (p = 0.58, Wilcoxon ranksum test). These analyses are now included as a sentence in the Results.

We agree with the reviewer that higher spontaneous activity in non-preferred spines may help explain the mitochondrial relationship we observe. However, our current dataset does not have sufficiently long recordings to measure spontaneous synaptic activity. Further, when considering a non-anesthetized preparation, spontaneous activity is highly dependent on brain state and an animal’s self-driven brain activity, which all must be experimentally controlled or measured to accurately address this.

1. In addition, the information about the orientation selectivity of the soma is also missing. Do the four cells shown here all have similar level of orientation selectivity? Or some have relative weak orientation selectivity in the soma?

Yes, all 4 cells have a similar OSI (range = 0.4 – 0.57, mean = 0.46 +/ 0.08 s.d.). This has been added to the Results section.

1. This study focused on only a fraction of spines that are (1) responsive (2) osi > 0.1. However, in theory energy consumption is also related to non-responsive spines and spines with weak orientation tuning. What is the percentage of tuned and untuned spines? What's the correlation of mitochondria volume and spine activity level for untuned spines? I also recommend including the non-responsive spines into the analysis. For example, for each mitochondrion calculate the averaged overall activity of spines within certain distance from the mitochondrion, including the non-responsive spines. I would predict there may be more active spines and higher overall spine activity of dentritic segments near a mitochondrion than segments far from a mitochondrion.

A majority of spines were tuned for orientation (~91%), although we specifically chose to only analyze data from spines with verifiable, independent calcium events. All analyses except those involving measurements of orientation preference use all dendritic spines (i.e. tuned and untuned). We have clarified this in the Results.

These other ‘silent’ (i.e. without resolvable visual activity) spines may significantly contribute to energy demands of a dendrite too, as our methods (GCaMP6s expression) likely only capture synaptic events driving Ca+2 influx through NMDA receptors or VGCCs. We expect that glutamate imaging (e.g. iGlusnfr) may open the door to additional analyses to fully characterize functional relationship between spines and mitochondria.

1. The correlation coefficient for mitochondria volume and difference in orientation preference is relatively low (r=0.3150). With such weak correlation, the explanatory power of this data is limited.

We agree that while the correlation is significant, it is not particularly strong. To better represent the noise surrounding this measurement, we performed a bootstrap correlation analysis, sampling with replacement (1 micron: mean r = 0.31 +/- 0.11 s.e., 5 micron: mean r = 0.02 +/- 0.10 s.e.). We now include this in the Results.

1. Why do the numbers of spines in different figures vary? For example, n=60 for 1micron in Figure 3, 54 in Figure 3c, 31 in Figure 4b, 51 in Figure 4e and so on.

We apologize for the lack of clarity. Each analysis presented different requirements of the data. For example, orientation preference was computed only for selective (OSI > 1) spines (Fig. 3c), but this requirement did not apply to comparisons with selectivity or response amplitude (Fig. 3d). Similarly, as stated in the Results and Methods, measurements of local heterogeneity require a minimum number of neighboring spines (n > 2), limiting the number of usable spines for analysis (Fig. 4). We have clarified this in the text.

1. In Figure 6a, the sample sizes of mito+ spines and mito- spines are extremely unbalanced, which affects the stat power of the analysis. I recommend performing a randomization test.

We thank the reviewer for this suggestion. We ran permutation tests to compare the similarity in mean values between equally sampled values from each distribution. These tests supported our original analysis and conclusions. We have added these tests to the Results.

1. Ca signals are approximations of electrical signals. How well are spinal calcium signals correlated to synaptic strength and local depolarization? This should be put into discussion.

There is unlikely a simple, direct relationship between spine calcium signal and synaptic strength or membrane depolarization, and this has never been addressed in vivo. Koester and Johnston (2005) performed paired recordings in slice and showed that single presynaptic action potentials producing successful transmission generate widely different calcium amplitudes (Fig. 3). Another study from Sobczyk, Scheuss, and Svoboda (2005) used two-photon glutamate uncaging on single spines and showed that micro-EPSC’s evoked are uncorrelated with the spine calcium signal amplitude. We have added a note about this in the discussion.

1. In Figure 4i, the negative correlation may depend on the 4 data points on the right side. How influential are those data points?

Spearman’s correlation coefficient analysis is robust to outliers and it is highly unlikely these datapoints are critical with our sample size (n > 100 spines).

1. Raw data of Ca responses were missing.

Some data has been published with the parent publication (Scholl et al., 2021). As spine imaging data is difficult to obtain and highly unique, we prefer to provide raw data directly upon reasonable request of the corresponding author.

1. What is the temporal frequency of the drifting grating? Was it fixed or the speed of the grating was fixed?

This was fixed to 4 Hz and this is now included in the Methods.

**Reviewer #2 (Recommendations For The Authors):**
1. Most of the measurements were based on the distance from the base of the spine neck, and "only on spines with measurable mitochondrial volume at each radius" were analyzed. To better understand the causality, it may also be interesting to have an analysis based on the distance from mitochondria. Would the result be different if the measurements are not 1µm / 5µm from spine but 1µm / 5µm from mitochondria? (e.g. total spine volume in 1µm / 5µm from mitochondria).

In fact, our first iteration of this study focused on exactly this metric: measuring the distance to nearest mitochondria. However, after lengthy discussions between the authors, we ultimately decided this metric was inferior to a volumetric one. Our decision was based on several factors: (1) distance to mitochondrion is ill defined (e.g. distance to the a mitochondrion center or nearest membrane edge?), (2) the total amount of mitochondrial volume within a dendritic shaft should allow the greatest amount of energetic support (e.g. more cristae for ATP production, greater capacity for calcium buffering), and (3) we would not account for the geometry of individual mitochondria or their placement near a spine (e.g. when 2 different mitochondria are next to the same spine) We have added further clarification of our reasoning to the Results.

Nonetheless, we present the reviewer some of our original analyses correlating distance to mitochondria (from the base of the spine and including the spine neck length):

**Author response image 1. sa4fig1:** Here, we examined the relationship to spine head volume, spine-soma orientation preference difference, and the local orientation preference heterogeneity. No relationship showed any significant correlation. Again, this may not be surprising given the drawbacks of measuring ‘distance to mitochondria’.

1. Is there a selection criterion for the spine for the analysis? Are filopodia spines excluded in the analysis?

Spines were analyzed regardless of structural classification; however, they were only analyzed if they had a synaptic density with synaptic vesicle accumulation. In our dataset (including those visualized in vivo and reconstructed from the EM volume) we observed no filopodia.

1. The result states that "56.8% of spines had no mitochondria volume within 1 μm and 12.1% of spines had none within 5 μm.". In other words, around 43% of spines had mitochondria within 1 μm. It would be interesting to show whether there is a correlation between mitochondria size and spine density.

We agree that this is an interesting measurement. It has been reported that mitochondrial unit length along the dendrite co-varies with linear synapse density in the neocortical distal dendrites of mice (Turner et al., 2022). This was specifically true in distal portions of dendrites more than 60 µm from the soma, because mitochondria volume increases as a function of distance roughly up to this point, then remains relatively constant beyond this distance.

To investigate this possibility, we calculated the local spine density around an individual spine and compared to the mitochondria volume within 1 or 5 µm. We found no evidence of a correlation between local spine density and the volume of mitochondria (1 µm: Spearman r = -0.07447, p = 0.2859; 5 µm: r = -0.04447, p = 0.3141). However, the majority of our measurements are more proximal than 60 µm (our median distance of all spines = 49.4 µm, max = 114 µm) and this may be one reason why observe no correlation.

1. In Figure 3B, the drifting grating directions are examined from 0 to 315 degrees in the experiment. However, in Figure 3C and 3D, the spine-soma difference of orientation preference was limited to 0 to 90 degree in the graph. Is the graph trimmed, or is there a cause that limits the spine-soma difference of orientation preference to 90?

Ferret visual cortical neurons are highly sensitive to grating direction and the responses are fit by a double Gaussian curve which estimates the ‘orientation preference’ (0-180 deg). We then calculated the absolute difference in orientation preference and wrapped that value in circular space so the maximum difference possible is 90 deg (e.g. 135 deg -> 45 deg).

1. In Figure 4D-F, how is the temporal correlation of calcium activity determined? Is it based on stimulated activity or basal activity? A brief explanation may be helpful to the readers. Also, scale bars could be added to Fig 4D.

Temporal correlation is computed as the signal correlation between 2 spines over the entire imaging session at that field of view. Specifically, we measured the Pearson correlation between each spine’s ∆F/F trace. To measure the local spatiotemporal correlation, we computed correlations between all neighboring spines within 5 microns and took the average of those values. We have clarified this in the Results section.

1. Figure 3C and Figure 4D displayed a significant correlation in 1µm range and such correlation drastically diminished once the criterion changed to 5µm range. It would be interesting to include the criterion of intermediate ranges. It would be interesting to see if there is a trend or tendency or if there is a "cut-off" limit.

We agree with the reviewer that the drastic change in the correlations between 1 and 5 µm range was surprising to see. While these volumetric measurements are time consuming, we returned to our data and measured an intermediate point of 3 µm. Investigating relationships reported in our study, we found no significant trends for spine-soma similarity (Spearman’s r = -0.011, p = 0.54) or local heterogeneity (Spearman’s r = 0.11, p = 0.23). This suggests that a potential ‘critical distance’ might be less than 3 µm; however, far more additional measurements and analyses would be needed to attempt to identify exactly what this distance is.

1. In Figure 5, it is shown that spines having mitochondrion in the head or neck are larger. However, only 10 spines are found with mitochondria inside. In the current dataset, are mitochondria abundantly found in large spines? Further analysis or justification would be informative to address this.

In our dataset, mitochondria were found in ~5% of all spines. Spines with mitochondria have a median volume of approximately 0.6 µm3, roughly twice as large as than those without mitochondria, as the reviewer suggests. In the entire population of spines without mitochondria, a volume of 0.6 µm3 represents roughly the 82nd percentile. In other words, of the total population of 157 spines without mitochondria, only 29 had equal or greater volume than the median spine with a mitochondrion. We believe this trend is clearly shown in Figure 5A and is supported by our analysis, including new permutation tests suggested by Reviewer 1.

**Reviewer #3 (Recommendations For The Authors):**
1. The authors state that their unsupervised method "quickly and accurately identified mitochondria," but the methods section only says that segmentations were proofread. Was every segmentation examined and judged to be accurate, or was only a subset of the 324 mitochondria checked?

After deep learning-based extraction, each mitochondrion segmentation was manually proofread. For each dendrite segment, this was ~10-20 mitochondria, so it did not take long to manually inspect and edit each mitochondrion segmentation.

1. The EM image of the mitochondrion in the spine head in Figure 2C is low resolution and does not apply to the bulk of the data. Images more representative of the analyzed data should be added to supplement the cartoons.

Our primary rationale for including this specific image was to show that the mitochondria located within spines are small, round, and to include a view of the synapse as well as the mitochondrion. We now include enlarge and additional EM images to Figure 1C.

1. The majority of spines did not have any mitochondria within a 1 micron radius and were excluded from the correlation analyses, so most of the conclusions are based on a minority of spines. It would be interesting to see comparisons between spines with and without nearby mitochondria. Correlations between the absolute distance to any mitochondrion, synapse size, and mismatch to soma orientation would be especially interesting.

The reviewer brings up a good point. It is true that many spines were excluded from our analysis based on the fact that they did not have nearby mitochondria within 1 or 5 µm (56.8% of spines had no mitochondria volume within 1 μm and 12.1% of spines had none within 5 μm). We compared the distributions of synapse size, mismatch to soma, and orientation selectivity of two groups of spines – those with at least some mitochondria within 1 µm (n = 65) versus spines without any mitochondria within 5 µm (n = 19).

We found no difference in the distributions between spine volume (1 µm: median = 0.29 µm3, IQR = 0.41 µm3; no mitochondria within 5 µm: median = 0.40 µm3, IQR = 0.37 µm3; p = 0.67) or PSD area (1 µm: median = 0.26 µm2, IQR = 0.33 µm2; no mitochondria within 5 µm: median = 0.31 µm2, IQR = 0.36 µm2; p = 0.49). For functional measures, we also saw no difference in orientation selectivity (1 µm: median = 0.29, IQR = 0.28; no mitochondria within 5 µm: median = 0.28, IQR = 0.15; p = 0.74) or mismatch to soma orientation (1 µm: median = 0.54 deg, IQR = 0.86 deg; no mitochondria within 5 µm: median = 0.46 deg, IQR = 0.47 deg; p = 0.75). We now include analyses in the Results.

We also looked at the absolute distances to mitochondria and did not find any significant relationships to spine head volume, spine-soma orientation preference difference, or the local orientation preference heterogeneity (see our response to reviewer #2 for more information).

1. In Figure 1A the mitochondria appear to be taking up a substantial fraction of the dendritic shaft diameter, even for distal dendrites. It would be useful to know the absolute diameter of the dendrites and mitochondria, given that this is not rodent data and there is no reference for either in the ferret.

We agree with the reviewer’s point, although we would like to remind the reviewer that these are basal dendrites of layer 2/3 cells. Basal dendrites tend to be thinner than apical branches. Interestingly, in some cases, the dendrite even swells to accommodate a mitochondrion. We did not incorporate this measurement in our study because it is not trivial; dendrite diameter is variable and dendrites are not perfect cylinders. Although we did not make precise measurements across our dendrites, the diameter is comparable to what has been seen in mouse cortex (Turner et al., 2022), roughly 500-1000 nm, but as small as 100 nm at some pinch points. In terms of mitochondria, many were roughly spherical or oblong, therefore the maximum diameters we report are roughly similar to, if not a bit larger than, those of the cross-sectional diameter.

1. As a rule, PSD area is correlated with spine volume, which makes the observation that spines with mitochondria have larger volume but not PSD area surprising. With n=10 it is difficult to draw conclusions, but it would be interesting to know the PSD area-to-volume ratio of other spines of the same volume and synapse size.

We were also somewhat surprised to see this, but exactly as the reviewer mentioned, we believe it to be a limitation of the sample size. The difference in volume was large enough to be detected despite a small sample size. We saw a trend towards larger synapses when spines have mitochondria (the median was approximately 60% larger), and we would expect with a larger comparison that PSD area would be significantly greater in spines with mitochondria.

We calculated the PSD area-to-spine head volume ratio for spines with or without mitochondria. Spines with mitochondria had a significantly lower ratio compared to those without (Mann-Whitney test, p = 0.0056, mito - = 0.78, n = 10; mito + = 0.53, n = 157). As the reviewer mentions, it is somewhat difficult to draw a conclusion from this, but it appears that the PSD does not scale with the increased spine head size.

**Author response image 2. sa4fig2:** 

The only way to definitively address this is to increase the sample size, which is becoming easier to achieve with the progression of volume EM imaging and analysis techniques in recent times. We look forward to addressing this in the future.

1. Nothing is made of the significant fact that these data come from the visual system of a carnivore, not a mouse. Consideration of differences in visual physiology between rodents and carnivores would be worthwhile to put the function of these dendrites in context.

We thank the reviewer for this consideration and have added text to the Discussion.